# Interpreting Graph Neural Networks for NLP with Differentiable Edge Masking

**Michael Sejr Schlichtkrull [1,2], Nicola De Cao [1,2], Ivan Titov [1,2]**
[1]University of Amsterdam, [2]University of Edinburgh
`m.s.schlichtkrull@uva.nl, n.decao@uva.nl, ititov@inf.ed.ac.uk`

## Abstract

Graph neural networks (GNNs) have become a popular approach to integrating structural inductive biases into NLP models. However, there has been little work on interpreting them, and specifically on understanding which parts of the graphs (e.g. syntactic trees or co-reference structures) contribute to a prediction. In this work, we introduce a *post-hoc* method for interpreting the predictions of GNNs which identifies unnecessary edges. Given a trained GNN model, we learn a simple classifier that, for every edge in every layer, predicts if that edge can be dropped. We demonstrate that such a classifier can be trained in a fully differentiable fashion, employing stochastic gates and encouraging sparsity through the expected $L_0$ norm. We use our technique as an attribution method to analyse GNN models for two tasks – question answering and semantic role labelling – providing insights into the information flow in these models. We show that we can drop a large proportion of edges without deteriorating the performance of the model, while we can analyse the remaining edges for interpreting model predictions.

## 1 Introduction

Graph Neural Networks (GNNs) have in recent years been shown to provide a scalable and highly performant means of incorporating linguistic information and other structural biases into NLP models. They have been applied to various kinds of representations (e.g., syntactic and semantic graphs, co-reference structures, knowledge bases linked to text, database schemas) and shown effective on a range of tasks, including relation extraction (Zhang et al., 2018; Zhu et al., 2019; Sun et al., 2019a; Guo et al., 2019), question answering (Sorokin & Gurevych, 2018; Sun et al., 2018; De Cao et al., 2019), syntactic and semantic parsing tasks (Marcheggiani & Titov, 2017; Bogin et al., 2019; Ji et al., 2019), summarisation (Fernandes et al., 2019), machine translation (Bastings et al., 2017) and abusive language detection in social networks (Mishra et al., 2019).

While GNNs often yield strong performance, such models are complex, and it can be difficult to understand the 'reasoning' behind their predictions. For NLP practitioners, it is highly desirable to know which linguistic information a given model encodes and how that encoding happens (Jumelet & Hupkes, 2018; Giulianelli et al., 2018; Goldberg, 2019). The difficulty in interpreting GNNs represents a barrier to such analysis. Furthermore, this opaqueness decreases user trust, impedes the discovery of harmful biases, and complicates error analysis (Kim, 2015; Ribeiro et al., 2016b; Sun et al., 2019b; Holstein et al., 2019). The latter is a particular issue for GNNs, where seemingly small implementation differences can make or break models (Zaheer et al., 2017; Xu et al., 2019). In this work, we focus on *post-hoc analysis* of GNNs. We are interested especially in developing a method for understanding how the GNN uses the input graph. As such, we seek to identify which edges in the graph the GNN relies on, and at which layer they are used. We formulate some desiderata for an interpretation method, seeking a technique that is:

1. able to identify relevant *paths* in the input graph, as paths are one of the most natural ways of presenting GNN reasoning patterns to users;
2. sufficiently *tractable* to be applicable to modern GNN-based NLP models;
3. as *faithful* (Jacovi & Goldberg, 2020) as possible, providing insights into how the model truly arrives at the prediction.

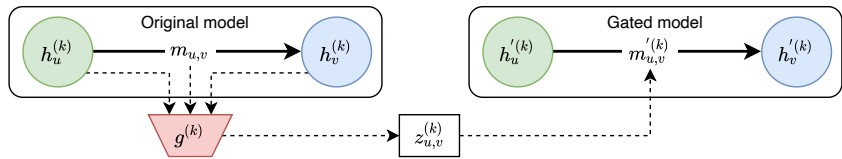

Figure 1: GRAPHMASK uses vertex hidden states and messages at layer $k$ (left) as input to a classifier $g$ that predicts a mask $z^{(\ell)}$. We use this to mask the messages of the $k$th layer and re-compute the forward pass with modified node states (right). The classifier $g$ is trained to mask as many hidden states as possible without changing the output of the gated model.

A simple way to perform interpretation is to use *erasure search* (Li et al., 2016; Feng et al., 2018), an approach wherein attribution happens by searching for a maximal subset of features that can be entirely removed without affecting model predictions. The removal guarantees that all information about the discarded features is ignored by the model. This contrasts with approaches which use heuristics to define feature importance, for example attention-based methods (Serrano & Smith, 2019; Jain & Wallace, 2019) or back-propagation techniques (Bach et al., 2015; Sundararajan et al., 2017). They do not guarantee that the model ignores low-scoring features, attracting criticism in recent years (Nie et al., 2018; Sixt et al., 2019; Jain & Wallace, 2019). The trust in erasure search is reflected in the literature through other methods motivated as approximations of erasure (Baehrens et al., 2010; Simonyan et al., 2014), or through new attribution techniques evaluated using erasure search as ground truth (Serrano & Smith, 2019; Jain & Wallace, 2019).

Applied to GNNs, erasure search would involve searching for the largest subgraph which can be completely discarded. Besides faithfulness considerations and conceptual simplicity, discrete attributions would also simplify the comparison of relevance between paths; this contrasts with continuous attribution to edges, where it is not straightforward to extract and visualise important paths. Furthermore, in contrast to techniques based on artificial gradients (Pope et al., 2019; Xie & Lu, 2019; Schwarzenberg et al., 2019), erasure search would provide implementation invariance (Sundararajan et al., 2017). This is important in NLP, as models commonly use highly parametrised decoders on top of GNNs (e.g., Koncel-Kedziorski et al. (2019)).

While arguably satisfying criteria (1) and (3) in our desiderata, erasure search unfortunately fails on tractability. In practical scenarios, it is infeasible, and even approximations, which remove one feature at a time (Zintgraf et al., 2017) and underestimate their contribution due to saturation (Shrikumar et al., 2017), remain prohibitively expensive.

Our GRAPHMASK aims at meeting the above desiderata by achieving the same benefits as erasure search in a scalable manner. That is, our method makes easily interpretable hard choices on whether to retain or discard edges such that discarded edges have no relevance to model predictions, while remaining tractable and model-agnostic (Ribeiro et al., 2016a). GRAPHMASK can be understood as a differentiable form of subset erasure, where, instead of finding an optimal subset to erase for every given example, we learn an erasure function which predicts for every edge $\langle u, v \rangle$ at every layer $k$ whether that connection should be retained. Given an example graph $\mathcal{G}$, our method returns for each layer $k$ a subgraph $\mathcal{G}_S^{(k)}$ such that we can faithfully claim that no edges outside $\mathcal{G}_S^{(k)}$ influence the predictions of the model. To enable gradient-based optimization for our erasure function, we rely on sparse stochastic gates (Louizos et al., 2018; Bastings et al., 2019).

In erasure search, optimisation happens individually for each example. This can result in a form of overfitting where even non-superfluous edges are aggressively pruned because a similar prediction could be made using an alternative smaller subgraph; we refer to this problem as *hindsight bias*. Because our interpretation method relies on a parametrised erasure function rather than an individual per-edge choice, we can address this issue by *amortising* parameter learning over a training dataset through a process similar to the readout bottleneck introduced in Schulz et al. (2020). In other words, the decision to drop or keep an edge is made based on the information available in the network (i.e., representation of the graph nodes) without having access to the final prediction (or to the gold standard). As we demonstrate in Section 4, this strategy avoids hindsight bias.

**Contributions**    Our contributions are as follows:

- We present a novel interpretation method for GNNs, applicable potentially to any end-to-end neural model which has a GNN as a component.[1]
- We demonstrate using artificial data the shortcomings of the closest existing method, and show how our method addresses those shortcomings and improves faithfulness.
- We use GRAPHMASK to analyse GNN models for two NLP tasks: semantic role labeling (Marcheggiani & Titov, 2017) and multi-hop question answering (De Cao et al., 2019).

## 2    RELATED WORK

Several recent papers have focused on developing interpretability techniques for GNNs. The closest to ours is GNNExplainer (Ying et al., 2019), wherein a soft erasure function for edges is learned individually for each example. Unlike our method (and erasure search), GNNExplainer cannot as such guarantee that gated edges do not affect predictions. Furthermore, as we show in our experiments (Section 4), separate optimisation for each example results in hindsight bias and compromises faithfulness. Pope et al. (2019); Xie & Lu (2019) explore gradient-based methods, including gradient heatmaps, Grad-CAM, and Excitation Backpropagation. Similarly, Schwarzenberg et al. (2019); Baldassarre & Azizpour (2019); Schnake et al. (2020) apply Layerwise Relevance Propagation (Bach et al., 2015) to the GNN setting. These methods represent an alternative to GRAPHMASK, but as we have noted their faithfulness is questionable (Nie et al., 2018; Sixt et al., 2019; Jain & Wallace, 2019), and the lack of implementation invariance (Sundararajan et al., 2017) is problematic (see Appendix H). Furthermore, significant engineering is still required to develop these techniques for certain GNNs, e.g. networks with attention as the aggregation function (Veličković et al., 2018).

Another popular approach is to treat attention or gate scores as a measure of importance (Serrano & Smith, 2019). However, even leaving questionable faithfulness (Jain & Wallace, 2019) aside, many GNNs use neither gates nor attention. For those that do (Marcheggiani & Titov, 2017; Veličković et al., 2018; Neil et al., 2018; Xie & Grossman, 2018), such scores are, as we demonstrate in Section 6, not necessarily informative, as gates can function to scale rather than filter messages.

Outside of graph-specific methods, one line of research involves decomposing the output into a part attributed to a specific subset of features and a part attributed to the remaining features (Shapley, 1953; Murdoch et al., 2019; Singh et al., 2019; Jin et al., 2020). For GNNs, the computational cost for realistic use cases (e.g. the thousands of edges per example in De Cao et al. (2019)) is prohibitive. LIME (Ribeiro et al., 2016b) like us relies on a trained erasure model, but interprets local models in place of global models. Local models cannot trivially identify useful paths or long-distance dependent pairs of edges, and as also pointed out in Ying et al. (2019) LIME cannot be easily applied for large general graphs. Similarly, it is unclear how to apply integrated gradients (Sundararajan et al., 2017) to retrieve relevant paths, especially for deep GNNs operating in large graphs.

Masking messages in GRAPHMASK can be equivalently thought of as adding a certain type of noise to these messages. Therefore, GRAPHMASK can be categorised as belonging to the recently introduced class of perturbation-based methods (Guan et al., 2019; Taghanaki et al., 2019; Schulz et al., 2020) which equate feature importance with sensitivity of the prediction to the perturbations of that feature. The closest to our model is Schulz et al. (2020), wherein the authors like us apply a secondary, trained model to predict the relevancy of a feature in a given layer. Unlike us, this trained model has 'look-ahead', i.e. access to layers above the studied layer, making their model vulnerable to hindsight bias. Their approach uses soft gates on individual hidden state dimension to interpolate between hidden states, Gaussian noise in order to detect important features for CNNs on an image processing task, and makes independent Gaussian assumptions on the features to derive their objective. We adapted their method to GNNs and used it as a baseline in our experiments.

In our very recent work (De Cao et al., 2020) we have introduced a similar differentiable masking approach to post-hoc analysis for transformers. We used sparse stochastic gates and $L_0$ regularisation to determine which input tokens can be dropped, conditioning on various hidden layers. Concurrently to this paper, Luo et al. (2020) have also developed an interpretability technique for GNNs relying on differentiable edge masking. Their approach uses a mutual information objective like GNNExplainer, along with local binary concrete classifiers as in GRAPHMASK.

---

[1]Source code available at `https://github.com/MichSchli/GraphMask`.

## 3 METHOD

### 3.1 GRAPH NEURAL NETWORKS

A Graph Neural Network is a layered architecture which takes an input graph $\mathcal{G} = \langle \mathcal{V}, \mathcal{E} \rangle$ (i.e., nodes and edges) to produce a prediction. At every layer $k$, a GNN computes a node representation $h_u^{(k)}$ for each node $u \in \mathcal{V}$ based on representations of nodes from the previous layer. At the bottom layer, vertices are assigned an initial embedding $h_u^{(0)}$ – e.g. GloVE embeddings, or the hidden states of an LSTM. For layers $k > 0$, a GNN can be defined through a message function $M$ and an aggregation function $A$ such that for the $k$-th layer:

$$m_{u,v}^{(k)} = M^{(k)}\left(h_u^{(k-1)}, h_v^{(k-1)}, r_{u,v}\right) \quad (1) \qquad h_v^{(k)} = A^{(k)}\left(\left\{m_{u,v}^{(k)} : u \in \mathcal{N}(v)\right\}\right), \quad (2)$$

where $r_{u,v}$ indicates the relation type between nodes $u$ and $v$, and $\mathcal{N}(v)$ the set of neighbour nodes of $v$. Typical implementations of GNNs rely on either mean-, sum-, or max-pooling for aggregation.

### 3.2 GRAPHMASK

Our goal is to detect which edges $(u, v)$ at layer $k$ can be ignored without affecting model predictions. We refer to these edges and the corresponding messages $m_{u,v}^{(k)}$ as *superfluous*. GNNs can be highly sensitive to changes in the graph structure. A GNN trained on graphs where all vertices $v$ have degree $d(v) \gg n$ for some integer $n$ may become unstable if applied to a graph where some vertices have degree $d(v) \ll n$. Hence, dropping edges without affecting predictions can be difficult. Nevertheless, many edges in that graph may be superfluous for all purposes other than normalization. Therefore, it is not enough to search for edges which can be *dropped* – instead, we search for edges which, through a binary choice $z_{u,v}^{(k)} \in \{0, 1\}$, can be replaced with a learned baseline $b^{(k)}$:

$$\widetilde{m}_{u,v}^{(k)} = z_{u,v}^{(k)} \cdot m_{u,v}^{(k)} + b^{(k)} \cdot (1 - z_{u,v}^{(k)}) . \tag{3}$$

Conceptually, the search for a subset that generates the same prediction can be understood as a form of subset erasure (Li et al., 2016; Feng et al., 2018). Unfortunately, erasure breaks with the principles we proposed in Section 1 in two important ways. First, since it involves searching over all the possible candidates that could be dropped, it is not *tractable*. Second, since the search happens individually for each example, there is a danger of *hindsight bias*. That is, the search algorithm finds a minimal set of features that could produce the given prediction, but which is not *faithful* to how the model originally behaved (as confirmed in our experiments, Section 4). To overcome those issues, we compute $z_{u,v}^{(k)}$ through a simple function, learned once for every task across data points:

$$z_{u,v}^{(k)} = g_\pi(h_u^{(k)}, h_v^{(k)}, m_{u,v}^{(k)}) , \tag{4}$$

where $\pi$ denotes the parameters of $g$, which is implemented as a single-layer neural network (see Appendix A for the architecture).

Instead of selecting gate values $z_{u,v}^{(k)}$ individually for each prediction, the parameters $\pi$ are trained on multiple datapoints, and used to explain predictions for examples unseen in the training phase. Moreover, each $z_{u,v}^{(k)}$ is computed relying only on information also available to the original model when computing the corresponding GNN message (i.e. states of nodes at layer $k$, $h_u^{(k)}$ and $h_v^{(k)}$). As such, the explainer is not provided with a look-ahead.[2] These two aspects, by design, work to prevent hindsight bias. We refer to this strategy as *amortisation*. The alternative to amortisation is to choose the parameters $\pi$ independently for each gate, without any parameter sharing across gates. In that case, optimisation would be performed directly on the analysed (i.e. test) examples. We refer to this strategy as the *non-amortized* version of GraphMask.[3] We will show in Section 4 that this version of GRAPHMASK, unlike the amortized approach, is susceptible to hindsight bias.

---

[2]The readout function in Schulz et al. (2020) violates this constraint.

[3]It would be wasteful to use a neural network $g_\pi(h_u^{(k)}, h_v^{(k)})$ in the non-amortized case and train it on a single example. Instead, we directly optimize the parameters of our stochastic relaxation, Hard Concrete, discussed in 3.3.

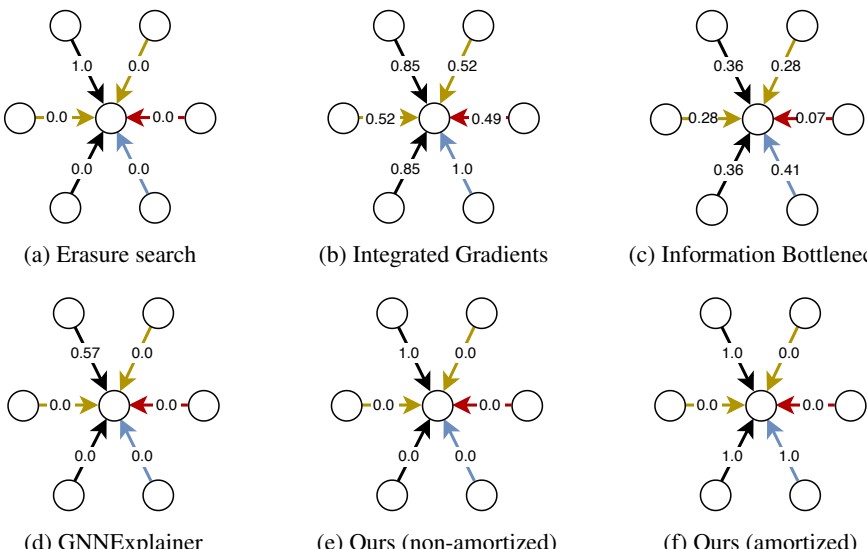

Figure 2: Toy example: a model predicts whether there are more black edges ($\rightarrow$) than blue edges. ($\rightarrow$). Erasure search, GNNExplainer, and non-amortized GRAPHMASK overfit by retaining only a single black edge (top left). Integrated gradients and the information bottleneck approach give unsatisfying results as all edges have attribution. Only amortized GRAPHMASK correctly assigns attribution to and only to black and blue edges.

After $g$ is trained, to analyse a data point with GRAPHMASK, we first execute the original model over that data point to obtain $h_u^{(k)}$, $h_v^{(k)}$, and $m_{u,v}^{(k)}$. We then compute gates for every edge at every layer, and execute a sparsified version of the model as shown in Figure 1. For the first layer, the messages of the original model are gated according to Equation 3. For subsequent layers, we aggregate the masked messages using Equation 2 to obtain vertex embeddings $h_v'^{(k)}$, which we then use to obtain the next set of masked messages. Note that the only learned parameters of GRAPHMASK are the parameters $\pi$ of the erasure function and the learned baseline vectors $b^{(1)}, \ldots, b^{(k)}$ – the parameters of the original model are kept constant. As long as the prediction relying on the sparsified graph is the same as when using the original one, we can interpret masked messages as superfluous.

## 3.3 PARAMETER ESTIMATION

Given a GNN $f$ of $L$ layers, a graph $\mathcal{G}$, and input embeddings $\mathcal{X}$ (e.g., initial node vectors or additional inputs), our task is to identify a set $\mathcal{G}_S = \{\mathcal{G}_S^{(1)}, \ldots, \mathcal{G}_S^{(L)}\}$ of informative sub-graphs such that $\mathcal{G}_S^{(k)} \subseteq \mathcal{G} \ \forall k \in 1, \ldots, L$. We search for a graph with the minimal number of edges while maintaining $f(\mathcal{G}_S, \mathcal{X}) \approx f(\mathcal{G}, \mathcal{X})$.[4] We can cast this, quite naturally, in the language of constrained optimization and employ a method that enables gradient descent such as Lagrangian relaxation. In general, however, it is not possible to guarantee equality between $f(\mathcal{G}, \mathcal{X})$ and $f(\mathcal{G}_S, \mathcal{X})$ since $f$ is a smooth function, and as therefore a minimal change in its input cannot produce the exact same output. As such, we introduce i) a divergence $\mathrm{D}_\star[f(\mathcal{G}, \mathcal{X}) \| f(\mathcal{G}_S, \mathcal{X})]$ to measure how much the two outputs differ, and ii) a tolerance level $\beta \in \mathbb{R}_{>0}$ within which differences are regarded as acceptable. The choice of $\mathrm{D}_\star$ depends on the structure of the output of the original model. A practical way to minimize the number of non-zeros predicted by $g$ is minimizing the $L_0$ 'norm' (i.e., the total number of edges that are not masked). Hence, formally, we define our objective over a dataset $\mathcal{D}$ as

$$\max_\lambda \min_{\pi,b} \sum_{\mathcal{G},\mathcal{X} \in \mathcal{D}} \left( \sum_{k=1}^{L} \sum_{(u,v) \in \mathcal{E}} \mathbf{1}_{[\mathbb{R} \neq 0]}(z_{u,v}^{(k)}) \right) + \lambda \left( \mathrm{D}_\star[f(\mathcal{G}, \mathcal{X}) \| f(\mathcal{G}_S, \mathcal{X})] - \beta \right) , \quad (5)$$

---

[4]With $f(\mathcal{G}_S, \mathcal{X})$ we denote a forward pass where for each layer the graph may vary, where for $f(\mathcal{G}, \mathcal{X})$ the graph $\mathcal{G}$ is the same across layers.

| Method | Prec. | Recall | $F_1$ |
|---|---|---|---|
| Erasure search* | 100.0 | 16.7 | 28.6 |
| Integrated Gradients | 88.3 | 93.5 | 90.8 |
| Information Bottleneck | 55.3 | 51.5 | 52.6 |
| GNNExplainer | 100.0 | 16.8 | 28.7 |
| Ours (non-amortized) | 96.7 | 26.2 | 41.2 |
| Ours (amortized) | 98.8 | 100.0 | **99.4** |

| Edge Type | $k = 0$ | $k = 1$ | $k = 2$ |
|---|---|---|---|
| MATCH (8.1%) | 9.4% | 11.1% | 8.9% |
| DOC-BASED (13.2%) | 5.9% | 17.7% | 10.7% |
| COREF (4.2%) | 4.4% | 0% | 0% |
| COMPLEMENT (73.5%) | 31.9% | 0% | 0% |
| Total (100%) | 51.6% | 28.8% | 19.6% |

Table 1: Comparison using the faithfulness gold standard on the toy task. *as in Li et al. (2016).

Table 2: Retained edges for De Cao et al.'s (2019) question answering GNN by layer ($k$) and type.

where $\mathbf{1}$ is the indicator function and $\lambda \in \mathbb{R}_{\geq 0}$ denotes the Lagrange multiplier.

Unfortunately, our objective is not differentiable. We cannot use gradient-based optimization since i) $L_0$ is discontinuous and has zero derivatives almost everywhere, and ii) outputting a binary value needs a discontinuous activation, e.g. the step function. A solution is to address the objective in expectation and employ either score function estimation i.e. REINFORCE (Williams, 1992), biased straight-through estimators (Maddison et al., 2017; Jang et al., 2017), or sparse relaxation (Louizos et al., 2018; Bastings et al., 2019). We choose the latter since it exhibits low variance compared to REINFORCE and is an unbiased estimator. We use the Hard Concrete distribution, a mixed discrete-continuous distribution on the closed interval $[0, 1]$. This distribution assigns a non-zero probability to exact zeroes. At the same time, it also admits continuous outcomes in the unit interval, for which an unbiased and low variance gradient can be computed via the *reparameterization trick* (Kingma & Welling, 2014). We refer to Louizos et al. (2018) for details. Attribution scores correspond to the expectation of sampling non-zero masks, since any non-zero value can leak information. In our experiments, GRAPHMASK converges to a distribution where scores in expectation assume near-binary values.

## 4 SYNTHETIC EXPERIMENT

We first apply GRAPHMASK in a setup where a clearly defined ground-truth attribution is known. As opposed to the real-world tasks we address in Sections 5 and 6, this allows for evaluation with respect to faithfulness. The task is defined as follows: a star graph $\mathcal{G}$ with a single centroid vertex $v_0$, leaf vertices $v_1, ..., v_n$, and edges $(v_1, v_0), ..., (v_n, v_0)$ is given such that every edge $(u, v)$ is assigned one of several colours $c_{u,v} \in C$. Then, given a query $\langle x, y \rangle \in C \times C$, the task is to predict whether the number of edges assigned $x$ is greater than the number of edges assigned $y$. We generate examples randomly with 6 to 12 leaves, and apply a simple one-layer R-GCN (Schlichtkrull et al., 2018) (see Appendix E for details). The trained model perfectly classifies every example. We know precisely which edges are useful for a given example – those which match the two colours being counted in that example. The GNN *must* count all instances of both to compute the maximum, and no other edges should affect the prediction. We define a gold standard for faithfulness on this basis: For $x > y$, all edges of type $x$ and $y$ should be retained, and all others should be discarded.

In Table 1, we compare GRAPHMASK to four baselines: erasure search (Li et al., 2016), integrated gradients (Sundararajan et al., 2017), an information bottleneck approach (Schulz et al., 2020), and GNNExplainer (Ying et al., 2019). Neither integrated gradients nor the information bottleneck approach were designed for graphs, and as such we adapt them for this setting (see Appendices F and G for details). Since GNNExplainer and Information Bottleneck do not make hard predictions, we define for both any gate $\sigma_i$ where $\sigma_i > t$ for some threshold $t$ as open, and closed otherwise. For integrated gradients we normalize attributions to the interval $[-1; 1]$, take the absolute value, and apply a threshold $t$. We select $t \in \{0.1, ..., 0.9\}$ to maximize $F_1$ score on validation data.

Only the amortized version of our method approximately replicates the gold standard. In fact, erasure search, GNNExplainer, and non-amortized GRAPHMASK recall only a fraction of the non-superfluous edges. Visually inspecting the scores assigned by various methods (Figure 2), we see that erasure search, GNNExplainer, and the non-amortized version of our method all exploit their training regime to reach the same low-penalty solution with perfect model performance, but which is not *faithful* to the original model behaviour. Since the task is to predict whether $x > y$, the model achieves a perfect score with only one edge of type $x$ retained. Conversely, for any $x \leq y$, the model

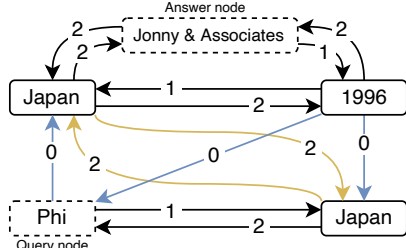
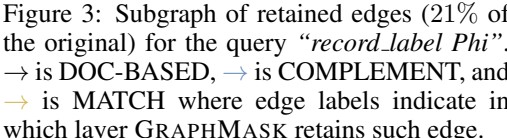

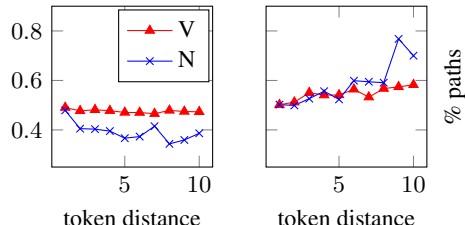

Figure 3: Subgraph of retained edges (21% of the original) for the query *"record_label Phi"*. → is DOC-BASED, → is COMPLEMENT, and → is MATCH where edge labels indicate in which layer GRAPHMASK retains such edge.

Figure 4: Percentage of paths used in predictions as a function of the distance between the predicate and the predicted role for the LSTM+GNN model (on the left) and the GNN only model (on the right).

achieves a perfect score with all edges dropped. Amortization prevents this type of overfitting to the objective. For integrated gradients, inspecting predictions shows that the scalar attribution scores vary greatly across examples with different numbers of edges. Hence, a single $t$ cannot be defined to always distinguish between useful and superfluous edges, even on this simple task.

## 5 QUESTION ANSWERING

We now apply GRAPHMASK (amortized) to analyze predictions for a real model. Due to the complexity, no human gold standard for attribution can be constructed in this setting (Jacovi & Goldberg, 2020). We choose the GNN-based model for multi-hop QA presented in De Cao et al. (2019), evaluated on WikiHop (Welbl et al., 2018). The task is, given a query sentence and a set of context documents, to find the entity within the context which best answers the query. Nodes in the GNN graph correspond to mentions of entities within the query and context, and four types of edges between those are introduced: string match (MATCH), document-level co-occurrence (DOC-BASED), coreference resolution (COREF), and, finally, the absence of any other edge (COMPLEMENT).

The model consists of a two-layer BiLSTM reading the query, and three layers of R-GCN (Schlichtkrull et al., 2018) with shared parameters. Node representations at the bottom layer are obtained by concatenating the query representation to embeddings for the mention in question. Here, we focus on their GloVe-based model. Finally, the mention representations are combined into entity representations through max-pooling.

GRAPHMASK replicates the performance of the original model with a performance change of $-0.4\%$ accuracy. 27% of edges are retained, with the majority occurring in the bottom layer (see Table 2). To ensure that the choice of superfluous edges is not just a consequence of the random seed, i.e. to verify the stability of our method, we compute Fleiss' Kappa scores between each individual measurement of $z_{u,v}^{(k)}$ across 5 different seeds. We find high agreement with $\kappa = 0.65$. Dropping just a random 25% of these retained edges greatly harms performance (see Appendix J).

For comparison, if we do not amortize to provide resilience against hindsight bias, the retained edges are different, with $0.4\%$ of retained edges in the bottom and $91.0\%$ in the top layer. Similarly, GNNExplainer and Integrated Gradients assign to the bottom layer, respectively, only $4.3\%$ and $11.3\%$ of their total attribution score. In contrast, dropping the bottom layer on all examples yields a much larger accuracy drop ($-26\%$) than any other layer (e.g., $-7\%$ for the top one). This suggests that these techniques do not produce faithful attributions. We provide more details in Appendix K.

In Table 2, we investigate which edge types are used across the three layers of the model. De Cao et al.'s (2019) ablation test suggested that COREF edges provide marginal benefit to the model; our analysis does not entirely agree. Investigating further, we see that only $2.3\%$ of the retained COREF edges overlap with MATCH edges (compared to $32.4\%$ for the entire dataset). In other words, the system relies on COREF edges only in harder cases not handled by the surface MATCH heuristic. The role COMPLEMENT edges play is interesting as well: this class represents the majority of non-superfluous edges in the bottom layer, but is always superfluous in subsequent layers. The model relies on an initial propagation-step across these edges, perhaps for an initial pooling of context.

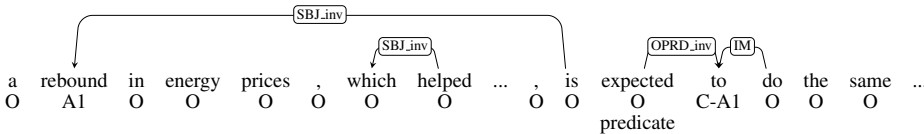

Figure 5: Example analysis on SRL from the GNN+LSTM model (superfluous arcs are excluded).

De Cao et al.'s (2019) model concatenates a representation of the query to every node in the graph before running GNN. As such, one might expect edges connecting mentions of the query entity to the rest of the graph to be superfluous. This, however, is not the case – at least one such edge is retained in $92.7\%$ of all cases, and in $84.1\%$ of cases in the bottom layer. We hypothesize that the model relies on GNN to see whether other mentions share a surface form or co-occur with mentions of the query entity, and, if not, how they otherwise connect to those. To investigate this, we measure the percentage of retained edges at each layer that occur on paths originating from query entities.

We find that the proportion of edges that occur on paths from mentions of the query increases drastically by layer, from $11.8\%$ at layer 0, to $42.7\%$ at layer 1, and culminating in $73.8\%$ in the top layer. A mention corresponding to the predicted answer is for $99.7\%$ of examples the target of *some* retained edge. However, the chance that the predicted entity is connected to the query ($72.1\%$) is near-identical to that of the average candidate entity ($69.2\%$). As such, the GNN is responsible not only for propagating evidence to the predicted answer through the graph, but also for propagating evidence to alternate candidates. The majority of paths take one of two forms – a COMPLEMENT edge followed by either a MATCH or a DOC-BASED edge ($22\%$), or a COMPLEMENT edge followed by two MATCH or DOC-BASED edges ($52\%$). MATCH and DOC-BASED edges in the bottom layer tend to represent one-hop paths rather than being the first edge on a longer path.

Relations used by De Cao et al. (2019) are symmetric (e.g., a coreference works in both directions). A distinct feature of the subgraphs retained by GRAPHMASK is that pairs of an edge and its inverse are *both* judged to be either superfluous or non-superfluous (individually in each layer). In Figure 3, this can be seen for the DOC-BASED edges in layer 2 between *Japan* and *Johnny & Associates*. Indeed, $49\%$, $98\%$ and $79\%$ of retained edges in, respectively, layers 0, 1 and 2 have their inverses also retained. In other words, 'undirected' message exchange between mentions, resulting in enriched mention representations, appears crucial.

| Type | Length | Retained edges | | | | |
| | | GNN-only | | | LSTM+GNN | |
| | | 0 | 1 | 2 | 0 | 1 |
|---|---|---|---|---|---|---|
| V | 1 (5755) | 0.01 | 0.99 | - | 0.01 | 0.99 |
| | 2 (1104) | 0.07 | 0.74 | 0.19 | 0.10 | 0.90 |
| | $\geq 3$ (10904) | 0.74 | 0.22 | 0.04 | 0.79 | 0.21 |
| N | 1 (3336) | 0.02 | 0.98 | - | 0.01 | 0.99 |
| | 2 (2935) | 0.30 | 0.25 | 0.45 | 0.89 | 0.11 |
| | $\geq 3$ (3251) | 0.56 | 0.32 | 0.12 | 0.73 | 0.27 |

Table 3: Percentages of paths with either 0, 1, or 2 edges retained, split by path length and predicate type, for the two models. For the LSTM+GNN model, at most one edge can be included per path as only a single GNN layer is employed.

## 6 SEMANTIC ROLE LABELING

We now turn to the GNN-based SRL system of Marcheggiani & Titov (2017). The task here is to identify arguments of a given predicate and assign them to semantic roles; see the labels below the sentence in Figure 5. Their GNN relies on automatically predicted syntactic dependency trees, allowing for information flow in both directions between syntactic dependents and their heads. We investigate both their best-performing model, which includes a BiLSTM and one layer of a GNN, and their GNN-only model.[5] For LSTM+GNN, the masked model has a minuscule performance change of $-0.62\%$ $F_1$ and retains only $4\%$ of messages. The GNN-only model has a similarly small performance change of $-0.79\%$ $F_1$ and retains $16\%$ of messages. We again compute Fleiss' Kappa scores between GRAPHMASK with 5 different seeds, finding a substantial agreement of respectively $\kappa = 0.79$ and $\kappa = 0.74$ for the full and GNN-only models.

---

[5]In Marcheggiani & Titov (2017), the best GNN-only model used three layers of GNN; with our reimplementation, a two-layer GNN performed better. Our reimplementation performed on par with the original.

The GNN, in this case, employs scalar, sigmoidal gates on every message. A naive method for interpretability could be to inspect their values. However, gates do not necessarily reflect the importance of individual messages; rather, they may provide scaling as a component in the model. On development data, the mean gate takes the value $0.16$, with a standard deviation of $0.07$. We evaluate the model with every message where the corresponding gate value is more than one $\sigma$ below the mean dropped, and find that performance decreases by $16.1\%$ $F_1$ score even though only $42\%$ of edges are removed. Thus, we see that these gates act as scaling rather than reflecting the contribution of each edge to the prediction (see also Appendix L for soft gate values for the example in Figure 5). This matches the intuition from Jain & Wallace (2019) that gates do not necessarily indicate attribution.

We first investigate which dependency types the GNN relies on. We summarise our finding in Figure 8 in Appendix I. The behaviour differs strongly for nominal and verbal predicates – NMOD dominates for nominals, whereas SBJ and OBJ play the largest roles for verbal predicates. This is unsurprising, because these edges often directly connect the predicate to the predicted roles. Even where this is not the case – see *rebound* in the example in Figure 5 – these edges connect predictions to tokens close to the predicate, easily reachable via the LSTM. Interestingly, several frequent relations (occurring in $> 10\%$ of examples) are entirely superfluous – these include P, NAME, CO-ORD, CV, CONJ, HYPH, SUFFIX, and POSTHON. For the LSTM-GNN model, we find that $88\%$ of retained edges point to predicted roles (e.g. *rebound*), and the remaining $12\%$ mostly point to arguments of other predicates in the same sentence (e.g. *which*).[6]

Marcheggiani & Titov's (2017) original findings suggest that the GNN is especially useful for predicting roles far removed from the predicate, where the LSTM struggles to propagate information. This could be accomplished by using paths in the graph; either relying on the entire path, or partially relying on the last several edges in the path. We plot in Figure 4 the percentage of paths from predicate to a predicted argument, such that a subpath (i.e. at least one edge) ending in the predicted argument was retained. For the LSTM+GNN model, we find that the reliance on paths decreases as the distance to the predicate increases, but only for nominal predicates. For the GNN-only model, we see the opposite: reliance on paths *increases* as the distance to the predicate increases. We investigate in Table 3 the proportion of edges retained on paths of varying length between the predicate and predicted roles. Practically all direct connections between the predicate and the roles are kept – this is unsurprising, as those edges are the most immediate indication of their syntactic relationships. Longer paths are often useful in both models, although at a lower rate for nominal predicates in the LSTM+GNN model. Our findings are consistent with the literature, where dependency paths connecting predicate and argument represent strong features for SRL (Johansson & Nugues, 2008; Roth & Lapata, 2016).

## 7 CONCLUSION

We introduced GRAPHMASK, a post-hoc interpretation method applicable to any GNN model. By learning end-to-end differentiable hard gates for every message and amortising over the training data, GRAPHMASK is faithful to the studied model, scalable to modern GNN models, and capable of identifying both how edges and paths influence predictions. We applied our method to analyse the predictions of two NLP models from the literature – an SRL model, and a QA model. GRAPHMASK uncovers which edge types these models rely on, and how they employ paths when making predictions. While these findings may be interesting per se, they also illustrate the types of analysis enabled by GRAPHMASK. Here we have focused on applications to NLP, where there is a strong demand for interpretability techniques applicable to graph-based models injecting linguistic and structural priors – we leave the application of our method to other domains for future work.

### 7.1 ACKNOWLEDGEMENT

The authors want to thank Benedek Rozemberczki, Elena Voita, Wilker Aziz, and Dieuwke Hupkes for helpful discussions. This project is supported by the Dutch Organization for Scientific Research (NWO) VIDI 639.022.518, SAP Innovation Center Network, and ERC Starting Grant BroadSem (678254).

---

[6] Note though that the GNN model of Marcheggiani & Titov (2017) 'knows' which predicate it needs to focus on, as its position is marked in the BiLSTM input.

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

## A    ERASURE FUNCTION ARCHITECTURE

We compute the parameters $\pi$ for the function erasure function $g_\pi$ defined in Equation 6 through a simple multilayer perceptron. We first derive a representation $q_{u,v}^{(k)}$ of an edge at layer $k$ simply through concatenation:

$$q_{u,v}^{(k)} = [h_u^{(k)}, h_v^{(k)}, m_{u,v}^{(k)}] \tag{6}$$

We then compute the scalar location parameters $\gamma_{u,v}^{(k)}$ for the hard concrete distribution based on $q_{u,v}^{(k)}$:

$$\gamma_{u,v}^{(k)} = W_2^{(k)}\text{ReLU}(\text{LN}(W_1^{(k)}q_{u,v}^{(k)})) \tag{7}$$

where LN represents Layer Normalization.

In addition to the formulation of GNN which we define in Equations 1 and 2, some implementations employ a faster – but less expressive – formulation, where aggregation is done through matrix multiplication between the vertex embeddings matrix $H^{(k)}$ and a (normalized, relation-specific) adjacency matrix $\hat{A}_r$ (Kipf & Welling, 2017; Schlichtkrull et al., 2018; De Cao et al., 2019):

$$H^{(k)} = \hat{A}_r H^{(k-1)} W^{(k)} \tag{8}$$

Applying the computation of $q_{u,v}^{(k)}$ from Equation 6 within that scheme would be prohibitively expensive, as $q_{u,v}^{(k)}$ would need to be computed for every possible combination of $u$ and $v$ rather than just those actually connected by edges. The complexity would as such rise to $O(V^2)$ rather than $O(V + E)$, which for large graphs can be problematic. To apply our method in such cases, we also develop a faster alternative computation of $\gamma_{u,v}^{(k)}$ based on a bilinear product. In this case rather than enumerating all possible messages, we rely purely on the source and target vertex embeddings $h_u^{(k)}$ and $h_v^{(k)}$. Taking inspiration from R-GCN (Schlichtkrull et al., 2018), we compute an alternative matrix-form $\hat{\gamma}^{(k)}$ as:

$$\hat{\gamma}^{(k)} = \hat{W}_r^{(k)}\text{ReLU}(\text{LN}(\hat{W}_1^{(k)} H^{(k)}))H^{(k)\top} \tag{9}$$

where $\hat{W}_r^{(k)}$ is unique to the relation $r$. We sample relation-specific matrix-form gates $\hat{Z}_r^{(k)}$, and apply these using an alternate – but equivalent – version of Equation 3 to derive a representation matrix $\widetilde{H}^{(k)}$ for the vertices in the masked model:

$$\sum_r (\hat{Z}_r^{(k)} \hat{A}_r) H^{(k-1)} W^{(k)} + ((J - \hat{Z}_r^{(k)}) \hat{A}_r) B^{(k)} \tag{10}$$

where $J$ represents the all-one matrix. In our experiments, we rely on the adjacency-list formulation for SRL in Section 6 and the adjacency-matrix formulation for QA in Section 5.

## B    THE HARD CONCRETE DISTRIBUTION

The Hard Concrete distribution assigns density to continuous outcomes in the open interval $(0, 1)$ and non-zero mass to exactly $0$ and exactly $1$. A particularly appealing property of this distribution is that sampling can be done via a differentiable reparameterization (Rezende et al., 2014; Kingma & Welling, 2014). In this way, the $L_0$ loss in Equation 5 becomes an expectation:

$$\sum_{k=1}^{L} \sum_{(u,v)\in\mathcal{E}} \mathbf{1}_{[\mathbb{R}\neq 0]}(z_{u,v}^{(k)}) = \sum_{k=1}^{L} \sum_{\langle u,v\rangle\in\mathcal{E}} \mathbb{E}_{p_\pi(z_{u,v}^{(k)}|\mathcal{G},\mathcal{X})} \left[ z_{u,v}^{(k)} \neq 0 \right] , \tag{11}$$

for which the gradient can be estimated via Monte Carlo sampling without the need for REINFORCE and without introducing biases.

**The distribution**    A stretched and rectified Binary Concrete (also known as Hard Concrete) distribution is obtained applying an affine transformation to the Binary Concrete distribution (Maddison et al., 2017; Jang et al., 2017) and rectifying its samples in the interval $[0, 1]$ (see Figure 6). A Binary Concrete is defined over the open interval $(0, 1)$ ($p_C$ in Figure 6a) and it is parameterised by a location parameter $\gamma \in \mathbb{R}$ and temperature parameter $\tau \in \mathbb{R}_{>0}$. The location acts as a logit and controls

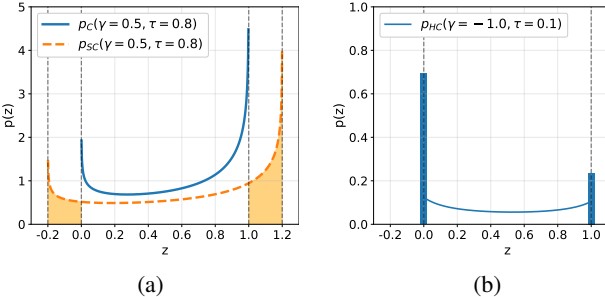

Figure 6: Binary Concrete distributions: (a) a Concrete $p_C$ and its stretched version $p_{SC}$; (b) a rectified and stretched (Hard) Concrete $p_{HC}$.

the probability mass skewing the distribution towards $0$ in case of negative location and towards $1$ in case of positive location. The temperature parameter controls the concentration of the distribution. The Binary Concrete is then stretched with an affine transformation extending its support to $(l, r)$ with $l \leq 0$ and $r \geq 1$ ($p_{SC}$ in Figure 6a). Finally, we obtain a Hard Concrete distribution rectifying samples in the interval $[0, 1]$. This corresponds to collapsing the probability mass over the interval $(l, 0]$ to $0$, and the mass over the interval $[1, r)$ to $1$ ($p_{HC}$ in Figure 6b). This induces a distribution over the close interval $[0, 1]$ with non-zero mass at $0$ and $1$. Samples are obtained according to

$$
\begin{aligned}
s &= \sigma\left(\left(\log u - \log(1 - u) + \gamma\right)/\tau\right) \\
z &= \min\left(1, \max\left(0, s \cdot (l - r) + r\right)\right)
\end{aligned}
\tag{12}
$$

where $\sigma$ is the Sigmoid function $\sigma(x) = (1 + e^{-x})^{-1}$ and $u \sim \mathcal{U}(0, 1)$. We point to the Appendix B of Louizos et al. (2018) for more information about the density of the resulting distribution and its cumulative density function.

In our experiments, we found a constant temperature $\tau = 1/3$ to work well. Message specific location parameters $\gamma_{u,v}^{(k)}$ are computed as specified in the previous section. We found it practical to shift the initial location using a bias $c = 2$, e.g. rather than directly using $\gamma_{u,v}^{(k)}$ in Equation 12 we substitute $\gamma_{u,v}^{(k)} + c$. This places the model in an initial state where all gates are open, which is essential for learning.

## C  TRAINING DETAILS

When training GRAPHMASK, we found it helpful to employ a regime wherein gates are progressively added to layers, starting from the top. For a model with $K$ layers, we begin by adding gates only for layer $k$, and train the parameters for these gates for $\delta$ iterations. We then add gates for the next layer $k - 1$, train all sets of gates for another $\delta$ iterations, and continue downwards in this manner. Optimising for sparsity under the performance constraint using the development set, we found the method to perform best with $\delta = 1$ for SRL, while the optimal setting for QA was $\delta = 3$.

We found it necessary to use separate optimizers and learning for the Lagrangian $\lambda$ parameter and for the parameters of GRAPHMASK. Thus, we employ Adam (Kingma & Ba, 2015) with initial learning rate $1e - 4$ for GRAPHMASK, and RMSProp (Tieleman & Hinton, 2012) with learning rate $1e - 2$ for $\lambda$. For the tolerance parameter $\beta$, we found $\beta = 0.03$ to perform well for all tasks.

We carried out all experiments on a single Titan X-GPU. As GRAPHMASK executes the model which it analyses, training- and run-time depends on the complexity of that model. At training time, GRAPHMASK requires a single forward pass to compute gate values, followed by a backward pass through the sparsified model. Thus, every iteration requires at most twice the computation time of an equivalent iteration using the investigated model.

# D  DATASETS

**SRL**  We used the English CoNLL-2009 shared task dataset (Hajič et al., 2009). This dataset contains 179.014 training predicates, 6390 validation predicates, and 10498 test predicates. The dataset can be accessed at `https://ufal.mff.cuni.cz/conll2009-st/`.

**QA**  For question answering, we used the WikiHop dataset (Welbl et al., 2018), and the preprocessing script from De Cao et al. (2019). See Table 4 for details. The dataset can be accessed at `https://qangaroo.cs.ucl.ac.uk/`.

|              | Min | Max   | Avg.  | Med. |
|--------------|-----|-------|-------|------|
| # candidates | 2   | 9     | 19.8  | 14   |
| # documents  | 3   | 63    | 13.7  | 11   |
| # tokens/doc.| 4   | 2,046 | 100.4 | 91   |

Table 4: WIKIHOP dataset statistics from Welbl et al. (2018): number of candidates and documents per sample and document length. Table taken from De Cao et al. (2019).

# E  SYNTHETIC TASK MODEL

For the synthetic task discussed in Section 4, we employ a model consisting of a one-layer R-GCN (Schlichtkrull et al., 2018). Vertex embeddings are initialized with the concatenation of a one-hot-encoding of $x$ and a one-hot-encoding of $y$. These are fed into an initial MLP with one hidden layer to construct zeroth-layer vertex embeddings $h_u^{(0)}$. For every leaf, messages are then computed as:

$$m_{u,v}^{(1)} = \text{ReLU}(W_{c_{u,v}} h_u^{(0)} + b_{c_{u,v}}) \tag{13}$$

Aggregation of messages is implemented as sum-pooling, and predictions are made from an MLP with one hidden layer computed from the embedding $h_{v_0}^{(1)}$ of the centroid. We use a dimensionality of 50 for R-GCN states, and a dimensionality of 100 for the MLP hidden states. The model is trained with Adam (Kingma & Ba, 2015), with an initial learning rate of $1e-4$.

# F  INTEGRATED GRADIENTS FOR GRAPHS

To apply integrated gradients to assign attributions to edges, we take the simplistic approach of defining a scalar variable $\hat{z}_{u,v}^k$ by which the message from $u$ to $v$ at layer $k$ is multiplied, and interpolate between $\hat{z}_{u,v}^k = 1$ and $\hat{z}_{u,v}^k = 0$. We then compute the relative attribution of $\hat{z}_{u,v}^k$, using 0 as a baseline; that is, we assume that the problem can be modelled as interpolating between edges being "fully present" and "fully absent" through "partially present" states. We note that it is nontrivial to extend this approach to multi-layer GNNs, since "partially present" edges in upper layers affect gradient flow and thus attribution to lower layers during interpolation. For this reason, information that has to travel through many edges – e.g., long-distance paths – is systematically underestimated in terms of importance. For the synthetic task where we rely on a single-layer GNN, we do not encounter this problem as no long-distance connections are possible; for real-world problems, this may not be the case (see e.g. our findings for QA in Appendix K). Furthermore, as we have noted in Section 3.2, the zero-vector may not be an appropriate baseline for general GNNs as it changes the degree statistics of the graph. This could harm the performance of integrated gradients (Sturmfels et al., 2020); however, as we have constructed our synthetic task such that the number of leaves and thus the degree of the centroid changes, a GNN which achieves a perfect score for this task must be robust with respect to changing degree statistics. To make binary predictions, we normalise attributions to the interval $[-1; 1]$, take the absolute value, and again apply a threshold $t \in \{0.1, ..., 0.9\}$ to determine useful and superfluous edges.

## G    INFORMATION BOTTLENECK FOR GRAPHS

A related attribution technique to ours is the information bottleneck approach proposed by Schulz et al. (2020). Their technique involves computing individual soft sigmoidal gates $\xi_{hi}$ for each dimension $i$ of the hidden state $h$ of a CNN to attribute importance. Instead of a learned baseline, gated vectors are replaced with samples from a Gaussian distribution. The mean and variance for this Gaussian are computed over all examples in the training dataset. To promote sparsity, the KL divergence between the dataset distribution and the distribution obtained by interpolating between that distribution and the observed value through the gate is used as regularisation.

In Section 4, we also include results for an adaptation of Schulz et al. (2020) to the problem of attributing importance to messages in a graph neural network. To apply the information bottleneck approach for our setting, we do the following. First, instead of individual gates $\xi_{mi}$ for each dimension of the message $m$, we use a single gate $\xi_m$. We use their Readout Bottleneck approach, which can be seen as a parallel to our amortisation strategy. We predict logits for each gate by conditioning on the source and target embeddings, as well as on the message itself, similar to how we compute parameters for GRAPHMASK (see Appendix A). This contrasts with the original approach of using $1x1$-convolutions over the depth dimension – conditioning on "downstream" messages in the GNN could cause hindsight bias. Training is done with the KL-divergence based loss introduced in Schulz et al. (2020). Finally, we compute the mean and variance of the Gaussian noise used as a baseline (and for the loss) in their approach using all messages in the same layer over the entire training dataset. We found using the entire training data to collect statistics to work better than collecting statistics individually per example.

## H    IMPLEMENTATION INVARIANCE FOR GNNS

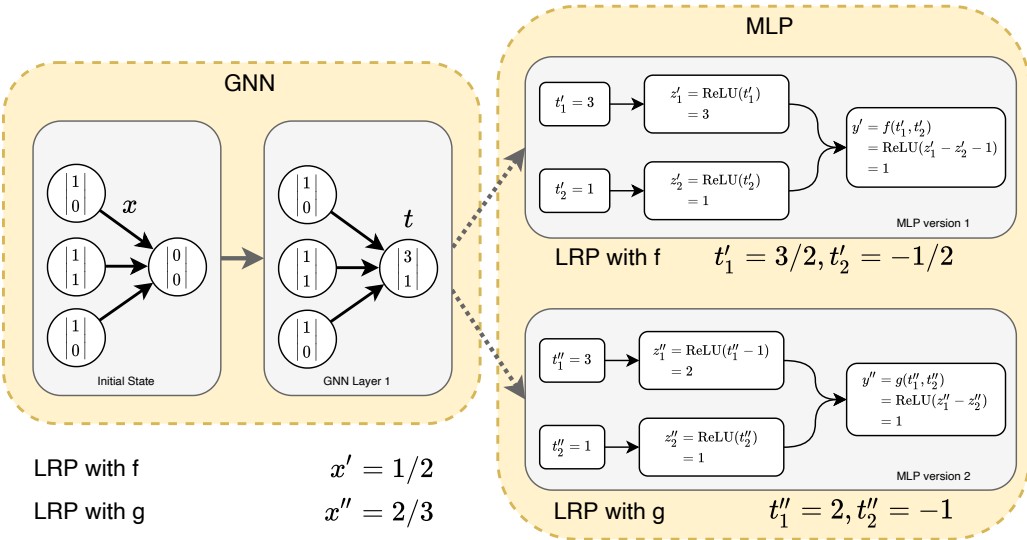

Figure 7: Attributions for two functionally equivalent networks. We give a graph as an input to a GNN (on the left) where $x$ is the edge from the top-right node to the central right node. The GNN update rule is simply aggregation with a sum over the neighbour nodes and no activation function. After one GNN layer we apply a MLP (on the right) which is implemented with for two functionally equivalent networks $f(t_1, t_2)$ and $g(t_1, t_2)$ (exactly the same as in the counterexample provided in Figure 7 in Sundararajan et al., 2017). Since LRP is not implementation invariant, it will produce two different attributions for the node $t$ (i.e., $t'$ and $t''$), and as a consequence of the propagation rule (Schwarzenberg et al., 2019), the attribution to $x$ will also be affected (i.e., $x'$ and $x''$).

## I    SRL Distribution over edge types for retained edges

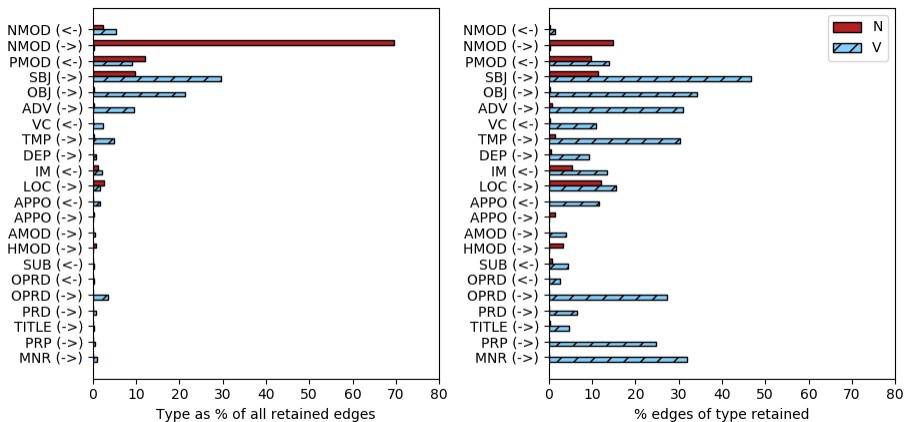

Figure 8: Distribution over edge types for retained edges (left) and probability of keeping each edge type (right); in both cases split by nominal (N) and verbal (V) predicates; edge types are a dependency function including computation directionality: flow from the head, (–>) or flow to the head (<–). Excludes edges that occur in less than 10 % cases, and edges judged superfluous in more than 99 % cases.

## J    Gradually dropping retained edges

| Retained edges | Acc. |
| --- | --- |
| 100% (Orig. model) | 59.0 |
| 27% (GRAPHMASK) | 58.6 |
| 20.25% | 55.2 |
| 13.5% | 52.8 |
| 6.25% | 47.7 |
| 0% | 45.2 |

(a) Question Answering

| Retained edges | F1 |
| --- | --- |
| 100% (Orig. model) | 87.1 |
| 4% (GRAPHMASK) | 86.6 |
| 3% | 83.1 |
| 2% | 74.3 |
| 1% | 68.9 |
| 0% | 63.8 |

(b) SRL: LSTM+GNN

| Retained edges | F1 |
| --- | --- |
| 100% (Orig. model) | 83.8 |
| 16% (GRAPHMASK) | 83.1 |
| 12% | 74.4 |
| 8% | 66.1 |
| 4% | 58.9 |
| 0% | 56.5 |

(c) SRL: GNN-Only

Table 5: Performance of the three real-world models using the original input graphs, using the subgraphs retained after masking with GRAPHMASK, and using only a randomly selected 0/25/50/75/100% of the edges retained after masking with GRAPHMASK. Dropping the edges marked superfluous by our technique does not impact performance; dropping the remaining edges, even if only a randomly selected 25% of them, significantly hurts the model.

## K    Baseline performance on Question Answering

Although we cannot directly measure and compare the faithfulness of different techniques on real tasks through a human-produced gold standard (Jacovi & Goldberg, 2020), we can identify clear pathologies in the attributions provided by both GNNExplorer and Integrated Gradients. An important clue is the level of attribution assigned by each technique to the individual layers of the GNN. In Figure 9, we plot for each layer of the Question Answering model the mean percentage of edges assigned specific attribution levels by each technique.

GNNExplainer and Integrated Gradients both assign low levels of attribution to the first two layers, relying primarily on the top layer. However, as we see in Table 6, dropping the bottom layer yields a much larger performance decrease (-26%) than dropping the top layer (-7%). This is at odds with the predicted attributions. For GNNExplainer, manual inspection reveals this to be a product of hindsight bias. Very specific configurations of top-layer edges adjacent to the predicted answer

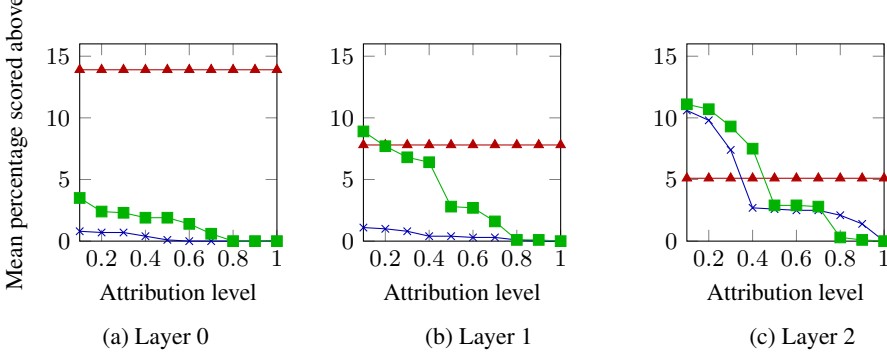

Figure 9: Mean percentage of messages assigned attribution scores above a certain level in the QA model of Section 5, separated by layer. We report scores for GNNExplainer (—×—), Integrated Gradients (—■—), and GRAPHMASK (—▲—).

(in most cases, retaining only edges where the predicted answer is the target) generates the same predictions as the original model. This mirrors a common pathology of erasure search on QA for textual data, where the answer span and nothing else is selected as an explanation (Feng et al., 2018).

For Integrated Gradients, the low scoring of the bottom layer is a result of long-distance information (e.g. information from edges and vertices far from the predicted answer, which must travel through half-open pseudo-gates in the other layers to reach the predicted answer) being systematically under-estimated as we discuss in Appendix F. This prevents meaningful comparisons of attribution scores between layers.

| Layers discarded | Accuracy |
| --- | --- |
| Full model | 59.0 |
| - layer 0 | 33.1 |
| - layer 1 | 41.6 |
| - layer 2 | 52.0 |

Table 6: Performance of the question answering model with all edges in each individual GNN layer dropped.

| Model | $k = 0$ | $k = 1$ | $k = 2$ |
| --- | --- | --- | --- |
| GNNExplainer | 4.3 | 11.9 | 83.8 |
| Integrated Gradients | 11.3 | 33.0 | 55.7 |
| GRAPHMASK | 51.6 | 28.8 | 19.6 |

Table 7: Mean percentage of the total attribution score allocated to each layer for the question answering model, according to GNNExplainer, Integrated Gradients, and GRAPHMASK.

Another approach is to compare the proportion of the total attribution score that different techniques assign to each layer; ideally, this should reflect the importance of that layer. In Table 7, we compute the mean percentage of the total score assigned to messages in each layer. As in Figure 9, we see GNNExplainer and Integrated Gradients assign low levels of attribution to the bottom layer, at odds with the empirical performance loss from excluding that layer. This again indicates that the baselines are unlikely to be faithful.

## L SRL EXAMPLE WITH SOFT GATES

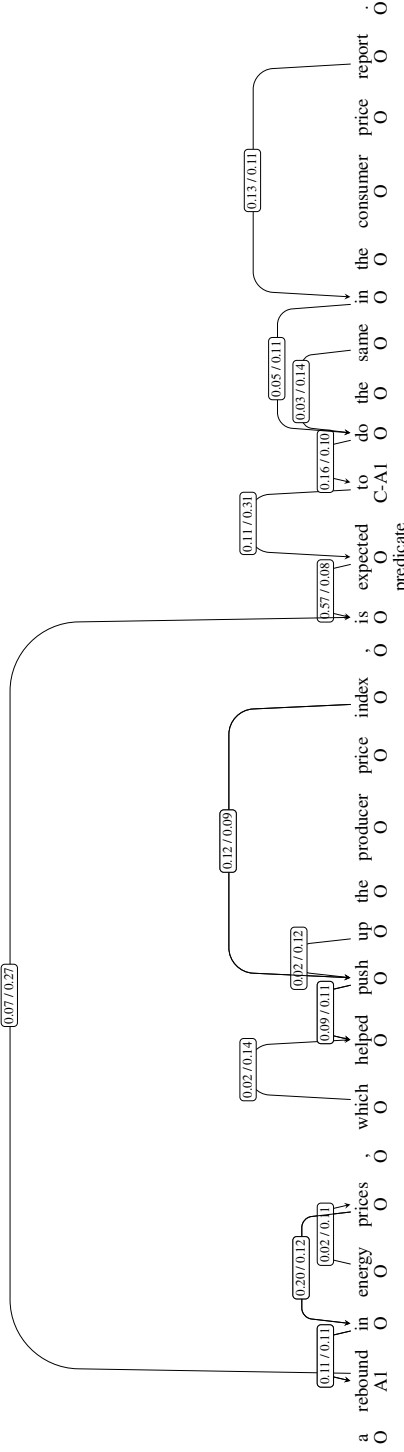

Figure 10: The example analysis from Figure 5, using the analysis heuristic where edges with soft gate values more than one standard deviation below the mean are discarded. Directions are combined into one arc.

