# OpenReview forum: "Interpreting Graph Neural Networks for NLP With Differentiable Edge Masking"
_ICLR.cc/2021/Conference — ICLR 2021 Spotlight_

### Official Review · AnonReviewer2 · 2020-10-27
**Explaining GNNs similar to prior work**

**Rating:** 7
**Confidence:** 4

**Review:**

The authors of the paper set out to design a method for explaining the behaviour of graph neural networks. Here the focus is on GNNs used in the context of natural language processing.

The authors argue that the following three properties of a method are desirable:

(1) The method should identify relevant paths across layers as paths are natural ways to present GNN reasoning patterns;
(2) Be tractable, that is, allow the computation of paths in an efficient manner;
(3) Be faithful.

The authors discuss existing post-hoc XAI approaches and the ways in which they do not meet at least one of the above desiderata.

(A friendly comment here: perhaps it would be better to tone down the language just a little bit. I’m not sure I’m convinced that paths are the most natural way to present GNN reasoning patterns, for instance. They are one possible way. Also, what is and isn’t faithful is still not really rigorously defined in the literature and your definition could be made a bit more specific.)

At this point, it would also make sense to define/make more explicit what you mean by “edge” and “paths” which you want to remove/retain to explain the behaviour of the GNN. Do you mean edge as in “edge between the input graph’s nodes in the computation graph induced by the GNN for one node”? Or do you mean “edge between neurons in the computation graph …”. Making this more concrete allows the user to follow your claims and reasoning better.

The proposed method belongs to the class of perturbation-based methods. It is very similar to prior work [1] and the authors should focus a bit more on similarities and differences in comparison to [1]. It seems the major difference is in the gating mechanism which is soft in [1] and in the application domain (image CNNs vs. GNNs). The lack of detailed discussion of [1] is especially problematic in the context of what the authors refer to as amortization. This seems to be crucial and makes a significant difference empirically yet is not discussed beyond a brief mention that this was introduced in [1]. The impression I currently have is that the paper is an application of [1] to GNNs.

One weakness of the proposed approach compared to prior work on explaining GNNs seems to be that it can only mask edges and not features. Is my understanding correct?

The experiments are well-designed and executed as far as I can tell. I am a proponent of the synthetic data experiments of the GNNExplainer and was happy to see them. It is interesting how well the integrated gradient method works compared to the GNNExplainer. I also like the NLP applications of GNNs and the corresponding experiments. Here, however, I would have wished to see a comparison again with integrated gradients and GNNExplainer. Is there a reason for not making such a comparison?

Overall I think this is a well-written and executed paper. The main problem is that it is very similar to prior work and doesn’t make enough of an effort to carve out the differences and similarities. The missing discussion of amortization is especially problematic, as it seems to be the decisive factor in the proposed method's performance.

[1] Karl Schulz, Leon Sixt, Federico Tombari, and Tim Landgraf. Restricting the flow: Information bottlenecks for attribution. In International Conference on Learning Representations, 2020.

---

> ### Author Response · Authors · 2020-11-16
> **Author response**
>
> Thank you for your comments!
>
> We appreciate your suggestions regarding the wording in the introduction and suggestions to spell out the relation to Schulz et al. (2020). That being said, we disagree that the methods are ‘very similar’.
>
> There are many differences besides the methods (besides 1. and 2. which you acknowledge):
> 1. We use HardConcrete which results in completely dropping the edge, whereas they add Gaussian noise
> 2.  Their input attributions are for CNNs and only on the image domain
> 3.  The objectives are different: ours is to maximize the number of dropped edges while maintaining the same performance as the original model, theirs is to minimize a linear combination of cross-entropy (w.r.t. gold standard) and an estimate of the mutual information.
> 4. They make strong assumptions about the feature distribution (independent Gaussians). While it may be reasonable for images, it is questionable in our case. If the estimates are off, their objective and hence attributions are not faithful
> 5. Their amortization relies on the states of the entire network. When applied to GNNs, it would, e.g., imply that decisions which edges to drop in hop 1 will be done based on the information available in the last hop. It results in the ‘hindsight bias’, similarly to not using any amortization. Instead, we do it incrementally, i.e. masks are predicted using the information available at the current hop.
>
> These differences are not arbitrary, they affect what conclusions one can make from analysis. E.g., in QA we see that COMPLEMENT edges are not useful in hops 2 and 3. Given that the computation is incremental (i.e. 5.) and the edges have been gated by exactly 0 (i.e. 1.), this implies that we can safely drop these edges completely and not propagate over them. There are no such guarantees for their method.
>
> Further, we implemented Schulz et al. (2020).  The best results we can obtain on the artificial test set is 43% F1. If we replace their amortization, which relies on information from “future” states (see 5. above), with ours, which relies on information available at the current hop, we obtain 52% F1, still much lower than 99% F1 with our method. See the new revision.
>
> To answer your other questions:
>
> > **One weakness of the proposed approach compared to prior work on explaining GNNs seems to be that it can only mask edges and not features.**
>
> It should be straightforward to generalize this framework to masking edge and node features (i.e. adding HardConcrete gates for these features), however, we have not experimented with it. For SRL, this is indeed interesting to see if it is the dependency labels which are important, or just the presence of a dependency relation.
>
> > **I would have wished to see a comparison again with integrated gradients and GNNExplainer [on real data].  Is there a reason for not making such a comparison?**
>
> We ran preliminary experiments using GNNExplainer on the QA task and the explanations were nonsensical. In most examples it retained only the subset of edges between text mentions referring to the predicted answer. This is a clear manifestation of hindsight bias. It is similar to the situation with QA models trained on SQuAD, where Erasure selects the answer span as an explanation [1].  For integrated gradients, the scores of edges in different layers will be on a different scale, making much of our multi-layer GNN analysis impossible (e.g., the ones in Table 3). We will compute the statistics and report it in a subsequent revision, hopefully before the end of the discussion period.
>
> > **I’m not sure I'm convinced that paths are the most natural way to present GNN reasoning patterns, for instance. They are one possible way.**
>
> We added a sentence to the introduction, clarifying that we are focusing on a specific problem, identifying which edges and paths are relevant.
>
> > **make more explicit what you mean by ‘edge’**
>
> We now refer to the graph as an ‘input graph’ when motivating the task, to distinguish it from the computation graph.
>
> [1] Shi Feng, Eric Wallace, Alvin Grissom II, Mohit Iyyer, Pedro Rodriguez, Jordan Boyd-Graber. Pathologies of Neural Models Make Interpretations Difficult. EMNLP, 2018

---

> > ### Comment · AnonReviewer2 · 2020-11-17
> > **What is amortization?**
> >
> > This was one of the central questions of my review. Did I miss your answer? Is it somewhere in the paper now?

---

> > > ### Author Response · Authors · 2020-11-17
> > > **Clarification regarding amortization**
> > >
> > > Our apologies for missing this part in our reply.  We use the term “amortization” to mean that we train a masking function (see Eq. (4)), parameterized by pi on the training set (see Eq. (5)) and use its prediction on the test set. This contrasts with "no amortization" (as, e.g., in GnnExplainer [1]), where each gate in every example has its own parameters. With no amortization, optimization is done for every test example independently.
> > >
> > > In our new revision, we clarify the difference between the amortized and non-amortized versions. See page 4,  starting with  Eq. (4) and ending with sentence "We refer to this strategy as the non-amortized version of GraphMask..."
> > >
> > > [1]: Ying, Zhitao, et al. "Gnnexplainer: Generating explanations for graph neural networks." Advances in neural information processing systems. 2019.

---

> > > > ### Comment · AnonReviewer2 · 2020-11-19
> > > > **Clarification regarding amortization**
> > > >
> > > > Dear authors,
> > > >
> > > > Thank you for the helpful responses to my questions. I am increasing my score by one point.
> > > >
> > > > I do not understand, however, your comment about the GNNExplainer providing "non-sensical" explanations and integrated gradients being on different scales for the NLP problems. This needs to be more rigorously discussed and the experimental results should be part of the paper. At least for one of the NLP problems where it is possible to argue that existing methods are not appropriate, both empirically and formally (e.g. what does it mean "the scores will be on different scales", can these scores be normalized,  why is this not the case for your method, etc.). If this can be included in an updated version, I will consider increasing my score further to accept.

---

> > > > > ### Author Response · Authors · 2020-11-20
> > > > > **Baseline results for question answering**
> > > > >
> > > > > We are very grateful for the time and effort you invest in evaluating our paper. We apologize for being a bit vague in that part of our response. It was taking some time to compute the exact statistics and make sure they are exactly comparable with each other.
> > > > >
> > > > > Now, we have included analysis of GNNExplainer and Integrated Gradients (IG) on the Question Answering task in the new revision of the paper (see paragraph 4 in Section 5 and especially Appendix K) and summarize the key points below.
> > > > >
> > > > > While it is generally hard to judge the faithfulness of an interpretation method on a real task, we can identify pathologies in the attribution provided by both GNNExplainer and Integrated Gradients.
> > > > >
> > > > > GNNExplainer mostly keeps only edges at the top layer of the GNN, driving the rest close to 0 (see Figure 9 and Table 7 in Appendix). We could interpret this as suggesting that the bottom layers are unimportant. However, removing the bottom GNN layer from the QA models yields a 26% drop in performance (Table 6), a much larger drop than from removing the top one (-7%). How is that possible? What GNNExplainer does is to find a very specific subgraph configuration yielding the same output as the original model. It only retains edges pointing to the vertices corresponding to the answer chosen by the original model (e.g., both mentions of "Japan" from Figure 3) in the top layer. Doing this, due to the structure of the QA model, guarantees that the answer will be "Japan". It is an artifact of optimization on the test set, i.e. a manifestation of what we refer to in the paper as a 'hindsight bias', and reminiscent of the pathologies discussed by Feng et al. (2018).
> > > > >
> > > > > The situation with IG is similar, though for a different reason (see discussion of IG's shortcomings in Appendices K and F). It also under-estimates the importance of bottom layers (also in Figure 9). Renormalization (within each layer) may help us answer questions about the relative importance of edges in each individual layer, but it is not the only question we may want to ask. We cannot answer such practically relevant questions as "which edges in the input graph are most important across the model?" (an edge type useful in the bottom layer will always be 'discriminated against'); "which layers are important?" or "which paths are important?".
> > > > >
> > > > > In contrast, the importance of layers determined by leave-one-layer-out experiments is consistent with GraphMask: most edges it retains are also in the bottom layer (Table 2).

---

### Official Review · AnonReviewer3 · 2020-10-27
**well-designed experiments and thorough analysis**

**Rating:** 7
**Confidence:** 4

**Review:**

*Summary of the paper*: This paper introduces a post-hoc method -- GraphMask, to interpret the prediction of GNNs. For each edge in the GNNs, the authors introduce a learnable hard gate which indicates whether this edge could be erased or not. The gates can be trained together with the model in a fully differentiable way. By analyzing the GNN models' behavior on two tasks -- question answering and semantic role labeling, the authors find that a large proportion of edges could be dropped without deteriorating the performance of the model. At the same time, the remaining edges could be used for interpreting model predictions.

*Strength of the paper*:
1. The idea to erase edges in GNNs is intuitive and straightforward, but how to do it properly is the problem. This paper introduces a simple yet viable method to deal with it. This kind of technique is potentially useful to analyze other models' behavior.

2. The experiments are well designed including both qualitative analysis, quantitative results, and reasonable ablation to show the effectiveness of their method. The thorough analysis conducted by the authors also contains useful insights of the model's behavior which is consistent with the literature.

3. The authors have done a thorough literature review and clearly state where does this paper stands.

*Comments*:

1. I really like the synthetic experiment designed in the paper which distinguishes the explanation proposed in the paper from others to be faithful. However, from my point of view, the task introduced here is too simple to say that such kind of faithfulness could be generalized to the real setting. I am curious if you further drop a small proportion of the remaining edges, how would it affect the performance?

2. Those model-dependent post-hoc explanations are interesting, I am curious how those explanations (remaining edges) would change if you have different encoders like BERT/Roberta. Could we treat it as a way to measure how much structural information is capture by the encoder? Also, other than the post-hoc explanation, do you have any idea of how we can further use such kinds of explanations to build a better/robust/faithful model?

*Reason for score*: Overall, I vote for accepting this paper. I like the idea of differentiable masking and the way the authors use to build faithful explanations for GNNs. The experimental results and analysis provide new insights to this area.

---

> ### Author Response · Authors · 2020-11-16
> **Author response**
>
> Thank you for your comments!
>
> To answer your questions:
>
> > **I am curious if you further drop a small proportion of the remaining edges, how would it affect the performance?**
>
> Partially dropping the retained edges and measuring performance is an interesting idea. We previously tried dropping all the remaining edges, which significantly harms performance (a drop of 13.4 points accuracy for QA, and respectively a 22.8 and 26.6 points F1-score for the SRL models with and without LSTM). Following your suggestion, we have also measured performance dropping a quarter/half/three quarters of the retained edges in addition to those already dropped by GraphMask. For question answering, the results are as follows:
>
> | Retained edges     | Accuracy |
> |--------------------|----------|
> | 100% (Orig. model) |     59.0 |
> |    27% (GraphMask) |     58.6 |
> |             20.25% |     55.2 |
> |              13.5% |     52.8 |
> |              6.25% |     47.7 |
> |                 0% |     45.2 |
>
> As can be seen, dropping the edges marked useful by our technique, even just 25% of them, significantly hurts the model. At the same time, the 73% of edges GraphMask marks superfluous can be dropped with only a 0.4% loss in performance. A similar trend can be observed for SRL -- see Appendix J of our revised submission.
>
> > **Could we treat [the change in retained edges] as a way to measure how much structural information is captured by the encoder?  What would change with BERT/Roberta?**
>
> Yes, using DiffMask as a way to probe for differences between encoders and reveal properties of contextualized embedding is an interesting direction, thank you for this suggestion. Studying the role of encoders was also the primary motivation for experiments on SRL, where we compared explanations when using the LSTM encoder vs using static embeddings. With LSTMs, the GNN component focuses primarily on long dependencies (Figure 4),  suggesting that short dependencies (and local context in general) are accurately captured by the LSTM itself.  The differences could be even more striking with contextualized encoders.
>
> > **How we can further use such kinds of explanations to build a better/robust/faithful model?**
>
> It is easy to imagine how such explanations can suggest ways to reduce or simplify models.  For example in our Question Answering experiments, COMPLEMENT edges appear not beneficial in hops 2 and 3 for Question Answering, so considerably smaller graphs could be used in these hops, reducing computation (see also our reply to AnonReviewer1). Regarding building more interpretable models, one option to try is to use the sparsification objective at training time, encouraging a simpler and more interpretable flow of computation.

---

### Official Review · AnonReviewer1 · 2020-10-29
**Novel interpretation method for graph neural networks with interesting application/discussions on multi-hop QA and SRL**

**Rating:** 7
**Confidence:** 3

**Review:**

Summary
This paper proposes an interpretation method for graph neural networks (GNNs) by learning parameterized, differentiable edge masks. The proposed method uses a single-layer NN classifier to predict  whether an edge can be dropped. It further uses L0 norm to encourage sparsity and amortizes parameter learning over a training dataset to avoid the hindsight bias. The method is experimented on a toy dataset (to demonstrate its faithfulness) and two real NLP tasks (multi-hop question answering and semantic role labeling). There are some interesting discussions based on the edge pruning results of these 2 tasks.

Strengths
- The paper proposes a novel interpretation method for GNNs. The method is technically interesting and well-motivated.
- The discussions for the QA and SRL models are thorough and interesting and provide useful insights. For example, the analysis shows that GNN-only models rely more on the paths for arguments farther away from the predicates, compared to LSTM-GNN models.
- The paper is well-written overall.

Weaknesses
- Some technical details are omitted. For example, “amortization” is only briefly mentioned in the last part of the introduction with a citation to Shultz et al., (2020). But this seems an important component of the model (shown in Section 4, Figure 2 and Table 1). It would be helpful to add more explanations/definitions so that the paper is more self-contained.
- The analysis and discussions for the QA and SRL tasks are very interesting. But they seem to be lacking actionable and generalizable conclusions.

---

> ### Author Response · Authors · 2020-11-16
> **Author response**
>
> Thank you for your comments!
>
> To answer your questions:
>
> > **“amortization’ is only briefly mentioned**
>
> As we have also noted in our response to AnonReviewer4, we have made an effort to clarify the term “amortization” in the revised version of the paper. We use the term to mean that we train a masking function (see Eq. (4)), parameterized by pi, on the training set (see Eq. (5)) and use its prediction on the test set. This contrasts with "no amortization" (as, e.g., in GnnExplainer [1]),  where each gate in every example has its own parameters. With no amortization, optimization is done for every test example independently. This is indeed a crucial step necessary to avoid hindsight bias, as we show in Section 4 of the paper. For that reason, we use the amortized model in our experiments on real data.
>
> > **The analysis and discussions .. are very interesting. But they seem to be lacking actionable and generalizable conclusions.**
>
> Our analysis on the two NLP tasks does have takeaways which can guide future model development -- for example, for QA, our observation that COMPLEMENT and COREF edges are not useful in hop 2 and 3 suggest that such edges could be excluded from computation to allow faster performance. For SRL, we find that certain dependency types are much more beneficial than others (see Appendix I). This suggests that increased performance on SRL could be obtained by choosing a dependency parser which performs well for these particular types, or even by retraining the dependency parser to prioritize these tasks.
>
> [1]: Ying, Zhitao, et al. "Gnnexplainer: Generating explanations for graph neural networks." Advances in neural information processing systems. 2019.

---

### Official Review · AnonReviewer4 · 2020-11-02
**I am waiting for the author's response.**

**Rating:** 6
**Confidence:** 4

**Review:**

Summary
This paper presents a new interpreting algorithm for understanding graph neural network (GNN) models for natural language processing (NLP) tasks. The main idea is to remove redundant edges for each layer of GNN after training the model. To find the redundant edges, the authors suggest minimizing the difference between the output of GNN with the original graph and inputs and the output of GNN with sub-graphs that remove some edges. The loss function forms the L0 norm with the Lagrange multiplier. The algorithm also learns baseline edge information that can replace the values of target edges. Experiment with synthetic datasets shows that suggested algorithms can find the important edges for the task rather than others that find only part of the important edges. Experiments on question answering and semantic role labeling with the real dataset, the suggested algorithm shows the percentage of retained edges and evidence that support the usefulness of the GNN algorithm for the task rather than LSTM.

The main strength of this paper is suggesting a sound and straightforward learning algorithm identify important edges for the task from the trained GNN model. Even the idea is simple; it can make people understand the GNN models better. I like to read the synthetic experiment section since it makes me admit to why the suggested algorithm much better than other algorithms. I also like to see the usefulness of the algorithm for the real NLP dataset.

What concerns me most is the lack of explanation about the amortized. The authors mentioned it in the Introduction section with Schulz et al. 2020 paper. But it is hard to imagine the amortized and non-amortized algorithms in the experiment sections. Following the result in Figure 2, amortization is a key factor in identifying the important edges. Could the authors explain it more about this on the suggested algorithm?



Overall, I admit that the suggested algorithm is a sound method to increase the interpretability of GNN models. But I have some questions that I want to listen to the author’s responses.


Questions
- How much drop the performance after masking? I know that the goal of masking is interpretability, not performance. But I am curious about it since if there are few drops, then we can identify informative sub-graphs by masking.
- Can we increase the percentage of masking nodes? Does it govern by the $\lambda$?
- Can we use the suggested algorithm for other NLP tasks that use GNN models such as [1, 2, 3]?

Typos
We investigate in Tables 3 and 3 on page 8

Reference
[1] https://www.aclweb.org/anthology/P18-1026/
[2] https://www.aaai.org/ocs/index.php/AAAI/AAAI18/paper/view/16329
[3] https://www.aclweb.org/anthology/P18-1149/

---

> ### Author Response · Authors · 2020-11-16
> **Author response**
>
> Thank you for your comments!
>
> We use the term “amortization” to mean that we train a masking function (see Eq. (4)), parameterized by pi on the training set (see Eq. (5)) and use its prediction on the test set. This contrasts with "no amortization" (as, e.g., in GnnExplainer [1]),  where each gate in every example has its own parameters. With no amortization, optimization is done for every test example independently. Using amortization is indeed a crucial step necessary to avoid ‘hindsight bias’, as we show in Section 4 of the paper. For that reason, we use the amortized model in our experiments on real data. We have made an effort to clarify this in the updated submission, please let us know if it addresses your concerns.
>
> To answer your other questions:
>
> > **How much drop the performance after masking?**
>
> The performance difference is negligible (see the first paragraphs of Section 5 and 6, respectively). For the synthetic task there is no difference. For Question Answering there is a 0.4% drop, while for SRL there is respectively a 0.62% and 0.79% drop for the LSTM+GNN and GNN-only models.  The fact that we can drop so many edges without hampering performance provides evidence that the dropped edges were indeed not beneficial.
>
> > **Can we increase the percentage of masking nodes? Does it govern by the λ?**
>
> We can drop more edges at the cost of performance by increasing the tolerance parameter β in equation 5. In our experiments we keep β very small to minimize the performance difference after masking.
>
> > **Can we use the suggested algorithm for other NLP tasks that use GNN models such as [1, 2, 3]?**
>
> Yes, we believe so. GraphMask makes no assumptions about the structure of the analysed model, other than that it uses a GNN to model graph-structured inputs.
>
> [1]: Ying, Zhitao, et al. "Gnnexplainer: Generating explanations for graph neural networks." Advances in neural information processing systems. 2019.

---

### Decision · Program_Chairs · 2021-01-07
**Final Decision**

**Decision:**

Accept (Spotlight)

**Comment:**

This paper proposes a method for  interpretable  graph neural networks.
The idea is intuitively well motivated: after training the model, discard spurious edges that are not critical to making predictions in the graph, and only retain salient edges.
Experiments on synthetic and real datasets show that the proposed method is effective at dropping only the edges that are not useful for the task at hand;  while leading to  only small performance degradation.  The paper is well-written. Overall, the paper brings together prior ideas in a useful way, and is well-executed.